# Synthesis and Biomedical Potential of Azapeptide Modulators of the Cluster of Differentiation 36 Receptor (CD36)

**DOI:** 10.3390/biomedicines8080241

**Published:** 2020-07-23

**Authors:** Caroline Proulx, Jinqiang Zhang, David Sabatino, Sylvain Chemtob, Huy Ong, William D. Lubell

**Affiliations:** 1Department of Chemistry, North Carolina State University, Raleigh, NC 27695, USA; cproulx@ncsu.edu; 2Innovative Drug Research Centre, Chongqing Key Laboratory of Natural Product Synthesis and Drug Research, School of Pharmaceutical Sciences, Chongqing University, Chongqing 401331, China; j.zhang1983@cqu.edu.cn; 3Department of Chemistry and Biochemistry, Seton Hall University, 400 South Orange Ave, South Orange, NJ 07079, USA; david.sabatino@shu.edu; 4Département d’Ophtalmologie, Université de Montréal, C.P. 6128, Succursale Centre-Ville, Montréal, QC H3C3J7, Canada; sylvain.chemtob@umontreal.ca; 5Faculté de Pharmacie, Université de Montréal, C.P. 6128, Succursale Centre-Ville, Montréal, QC H3C3J7, Canada; huy.ong@umontreal.ca; 6Département de Chimie, Université de Montréal, C.P. 6128, Succursale Centre-Ville, Montréal, QC H3C3J7, Canada

**Keywords:** semicarbazide, CD36, age-related macular degeneration, atherosclerosis, macrophage-driven inflammation, peptide

## Abstract

The innovative development of azapeptide analogues of growth hormone releasing peptide-6 (GHRP-6) has produced selective modulators of the cluster of differentiation 36 receptor (CD36). The azapeptide CD36 modulators curb macrophage-driven inflammation and mitigate atherosclerotic and angiogenic pathology. In macrophages activated with Toll-like receptor-2 heterodimer agonist, they reduced nitric oxide production and proinflammatory cytokine release. In a mouse choroidal explant microvascular sprouting model, they inhibited neovascularization. In murine models of cardiovascular injury, CD36-selective azapeptide modulators exhibited cardioprotective and anti-atherosclerotic effects. In subretinal inflammation models, they altered activated mononuclear phagocyte metabolism and decreased immune responses to alleviate subsequent inflammation-dependent neuronal injury associated with retinitis pigmentosa, diabetic retinopathy and age-related macular degeneration. The translation of GHRP-6 to potent and selective linear and cyclic azapeptide modulators of CD36 is outlined in this review which highlights the relevance of turn geometry for activity and the biomedical potential of prototypes for the beneficial treatment of a wide range of cardiovascular, metabolic and immunological disorders.

## 1. Introduction

In modern biomedical science, the transformation of biologically active peptides into potent and selective therapeutic prototypes is a challenge requiring an interplay of chemical synthesis and biological analysis. Over more than a decade of such collaborative research, unselective peptide hits exhibiting binding affinity for the cluster of differentiation 36 receptor (CD36) were transformed into potent and selective azapeptide leads with promising medicinal potential. In the absence of structural data for CD36 alone nor in complex with small molecules, lead optimization relied on structure–activity relationship data obtained from different assays designed to assess the pleiotropic effects of the receptor. During this research novel methods were conceived for the synthesis of linear and cyclic azapeptides. Certain azapeptides were shown to reduce inflammation by a mechanism featuring binding of CD36 and disruption of its interaction with the Toll-like receptor (TLR)-2/6 heterodimer to subsequently perturb downstream signaling and attenuate the pro-inflammatory cascade. In choroidal explant models, some azapeptides reduced neovascularization mediated by the TLR-2 heterodimer. Other CD36-selective azapeptides exhibited cardioprotective effects associated with an increase in circulating adiponectin levels. This review focuses on the chemistry leading to a better understanding of the topological requirements for binding and regulating CD36 activity towards the discovery of novel prototypes for treating cardiovascular, metabolic and immunological conditions. 

## 2. Discussion 

### 2.1. The Cluster of Differentiation 36 Receptor (CD36) Is a Medicinally Relevant Target 

CD36 is a transmembrane glycoprotein expressed on platelets, monocytes, macrophages, hepatocytes, endothelial and several other cell types [1,2,3]. CD36 is comprised of a single 472-amino acid chain cross-linked by three disulfide bonds and post-translationally modified by significant glycosylation and phosphorylation [4]. The extracellular loop of CD36 is relatively large, binds diverse circulating ligands and is flanked by two transmembrane spanning regions at the *N*- and *C*-termini with short cytosolic domains that are anchored into the membrane by four palmitoylation sites. Functioning as a class B scavenger receptor, CD36 is involved in the regulation of inflammatory processes, inhibition of microvascular angiogenesis, transport of oxidized lipoproteins and phospholipids, phagocytosis of microorganisms and cells, and autophagy [1,2,3,4,5,6,7]. CD36 binds to multiple endogenous ligands, including long chain fatty acids, the extracellular matrix protein thrombospondin 1 (TSP1), oxidative low-density lipoprotein (oxLDL), apoptotic cells and photoreceptor outer segments. Ligand engagement by CD36 has important implications in cardiovascular biology, including events leading to atherosclerosis, angiogenesis, diabetic retinopathy and age-related macular degeneration (AMD) [1,2,3,5,6,7].

As a translocator of long chain fatty acids, CD36 plays a critical role in cardiac metabolism. During ischemia-reperfusion injury, myocardial function is altered with increased uptake and oxidation of fatty acid at the expense of glucose resulting in contractile dysfunction. Myocardial damage can be mitigated, and cardiomyocyte function ameliorated by inhibition of fatty acid uptake and mitochondrial oxidation in the myocytes [8,9]. Myocardial ischemia and reperfusion activate the sympathoadrenal system and catecholamine-induced lipolysis causing increases in the availability of circulating levels of non-esterified fatty acids driving myocardial fatty acid uptake and oxidation, which may hinder glucose metabolism, a requirement for normal cardiac function [10]. Implicated in regulating myocardial non-esterified fatty acid uptake, CD36 regulates both circulating and myocardial levels of adiponectin, a major cardioprotective adipokine, that interplays with the transcription factor peroxisome proliferator-activated receptor-γ (PPARγ), which is a key regulator of adiponectin gene expression [11]. Adiponectin elicits antilipolytic effects and reduces circulating non-esterified fatty acids [12], which in turn regulates myocardial anti-apoptotic, antioxidant and metabolic functions. 

In addition to activity in cardiovascular biology, CD36 displays major roles in innate immunity, in eliciting reactive oxygen species production and in modulating metabolic homeostasis in immune cells [13]. As a co-receptor of the Toll-like receptor (TLR)-2/6 heterodimer assembly, CD36 regulates TLR-2-dependent macrophage-driven inflammation by sustaining TLR-2/6 signaling on the surface of membranes of mononuclear phagocytes [14,15,16]. The CD36-TLR-2/6 protein–protein interaction appears as a promising target for therapeutic intervention to treat retinal inflammation associated with accumulated mononuclear phagocyte activation, and photoreceptor degeneration neurotoxicity induced by photo-oxidative stress. In consideration of these pleiotropic biological effects, binding ligands that dissociated CD36 interaction with the TLR-2 heterodimer offer interesting therapeutic potential. 

### 2.2. Growth Hormone Releasing Peptide-6 (GHRP-6) Analogues as Lead CD36 Binding Ligands 

Several CD36 ligands have demonstrated the capacity to protect and preserve normal cardiac function [11,12,17]. Synthetic analogues of growth hormone releasing peptide-6 (GHRP-6, H-His-D-Trp-Ala-Trp-D-Phe-Lys-NH_2_, **1**, CD36 IC_50_ 1.82 μM) have shown promising cardioprotective effects by binding and regulating CD36 in a mouse model of myocardial ischemia and reperfusion [9]. For example, the GHRP-6 analogue EP80317 (Haic-D-Trp(2-Me)-D-Lys-Trp-D-Phe-Lys-NH_2_, **2**, CD36 IC_50_ 1.11 μM, Haic = 5-amino-1,2,4,5,6,7-hexahydroazepino[3,2,1-*hi*]indol-4-one-2-carboxylic acid) exhibited cardioprotective activity, reduced myocardial fatty acid uptake and prevented non-esterified fatty acid mobilization from adipose tissue [9]. The potential for ligands based on the GHRP-6 structure to target ischemic cardiomyopathy was however limited due to their lack of receptor selectivity, binding to both CD36 and the growth hormone secretagogue receptor (so-called ghrelin receptor, GHS-R1a). Certain azapeptide GHRP-6 analogues have surmounted this limitation, retaining high binding affinity for CD36 without exhibiting significant GHS-R1a binding affinity (*vide infra*) [18].

Topographical mapping of the binding hotspot domains of CD36 with a benzophenone alaninyl (Bpa) hexarelin derivative [Tyr-Bpa-Ala-His-D-(2-Me)Trp-Ala-Trp-D-Phe-Lys-NH_2_, **3**, CD36 IC_50_ 2.08 μM] revealed that the lysine-rich Asn^132^-Gln^177^ extracellular domain which interacted with GHRP-6 ligands overlapped with the oxLDL binding site [19]. Within this region, the positively-charged side chains of Lys^164^ and Lys^166^ have been found to directly contribute to oxLDL ligand binding to CD36 [20]. Considering the structures of GHRP-6 peptide ligands and the Lys-rich binding domain of CD36, aromatic–cation interactions were hypothesized to contribute to ligand-receptor binding affinity [21].

The binding affinity measurements (IC_50_ values) of peptide ligands for CD36 and the GHS-R1a receptor (GHS-R1a) were respectively determined by competitive binding assays using [^125^I]-radiolabeled photoactivatable Bpa-GHRP-6 derivative **3** as tracer and by a ghrelin binding assay with [^125^I]-ghrelin as radiotracer [18,21]. CD36 ligand binding affinity (EC_50_ and K_d_ values) has also been measured using surface plasmon resonance (SPR) analysis on a surface consisting of a peptide-based self-assembled monolayer (SAM) containing a recombinant His-tagged receptor [22]. Gold-surface-coated peptide monolayers consisting of 3-mercaptopropionyl-leucinyl-histidinyl-aspartyl-leucinyl-histidinyl-aspartic acid were functionalized with *N*′,*N*′-bis(carboxymethyl)-L-lysine using 1-ethyl-3-(3-dimethylaminopropyl)carbodiimide (EDC) and *N*-hydroxysuccinimide (HOSu) peptide-based coupling chemistry. The peptide-functionalized surface enabled immobilization of nitrilotriacetic acid for copper chelation and binding to recombinant histidine-tagged CD36, which could be reversed with imidazole treatment. The immobilization of CD36 onto the peptide monolayer provided a SPR means for screening binding ligands, such as GHRP-6 (**1**), EP80317 (**2**), and selected GHRP-6 azapeptides. The relative binding affinity measurements (EC_50_ values) of the peptide analogues correlated with CD36-dependent phosphorylation of sarcoma, src-kinases (Lyn/Fyn) in CD36 expressing J774 macrophages [5]. Moreover, the apparent binding affinity measurements (K_d_ values) were found to be consistent (1 μM–70 μM) with those derived from the previously established competitive binding assays [18,21]. The SPR biosensor platform for studying surface immobilized receptor–ligand binding interactions may provide additional insight into requirements for selective ligand binding affinity and downstream biological responses. 

### 2.3. Conception of Azapeptide GHRP-6 Ligands Exhibiting CD36-Selective Binding Affinity 

Exhibiting promising cardioprotective activity, the GHRP-6 peptides suffered from a lack of receptor selectivity. The parent peptide GHRP-6 (**1**) bound respectively to CD36 and the GHS-R1a receptor in the low micro- and nanomolar ranges (EC_50_ values, Table 1). Peptide **1** has been characterized to adopt a random coil conformation in solution based on circular dichroism spectroscopic analysis (*vide infra*) [18], in spite the central D-Trp-L-Ala-L-Trp-D-Phe sequence possessing alternating L- and D-amino acid pairs, which are commonly found at the central resides of β-turns [23]. Considering that a preferred turn geometry may be adopted on receptor binding, GHRP-6 analogues that discriminate between CD36 and the GHS-R1a receptor were pursued by employing semicarbazide residues as amino amide surrogates to favor such conformers and improve selective binding affinity. In azapeptides, semicarbazide residues can favor backbone β-turn geometry and improve metabolic stability [24]. Azapeptide GHRP-6 analogues served to elucidate structural requirements for selective CD36 binding affinity. Certain azapeptides retained low micromolar CD36 binding affinity, but lost significant (10^2^-10^4^-fold) binding affinity for the GHS-R1a receptor (Table 1).

Aza-GHRP-6 analogues were identified which bound effectively CD36 without any binding affinity for the GHS-R1a receptor. An aza-amino acid scan was initially performed in which each residue of the central D-Trp^2^-Ala^3^-Trp^4^ region of GHRP-6 was replaced by aza-residues [18]. Fifteen azapeptides were synthesized, in which the D-Trp^2^ and Trp^4^ residues were respectively substituted with aromatic aza-amino acids (azaPhe, azaTyr, azaBip, azahPhe and azaNal-1, Bip = *p*-phenyl-Phe, hPhe = homo-Phe, Nal = naphthylalanine), and the Ala^3^ residue was replaced with azaGly and azaLeu (Figure 1). Although azaTrp analogues were synthesized in solution, the indol-3-yl methyl side chain was labile under acidic conditions, such that azaGly peptides resulted from exposure of the azaTrp counterparts to the trifluoroacetic acid (TFA) conditions used for resin cleavage [25]. Examination of their respective binding affinities revealed that four of the fifteen azapeptide GHRP-6 analogues (e.g., **5**-**8**) retained micromolar binding affinity for CD36 but exhibited 10^2^-10^4^-fold losses in GHS-R1a receptor binding (Table 1).

A deeper understanding of the structural requirements for CD36 binding selectivity and affinity was obtained by performing an alanine scan on five azapeptides (**4**–**8**). In the Ala-scan of the GHRP-6 azapeptides, selective substitutions of L- and D-alanine residues throughout the sequence except the aza-residues enabled study of the importance of side chains on biological activity. So-called IRORI Kans™ technology was used to prepare twenty-two analogues by split-and-mix parallel combinatorial synthesis (*vide infra*). Six aza-GHRP-6 analogues were subsequently identified with improved CD36 selectivity and micromolar range binding affinity (Table 2). Notably, replacement of the *N*-terminal His by Ala resulted typically in a >10^4^-fold loss of binding affinity for the GHRS-R1a receptor without perturbing CD36 engagement (e.g., **10**–**12** and **15**). Moreover, respective substitutions of L- and D-Ala for the His^1^, Trp^4^ and D-Phe^5^ residues of [azaLeu^3^]GHRP-6 (**6**) provided analogues which maintained CD36 binding affinity with improved selectivity (e.g., **12**–**14**). Although micromolar affinity to both the CD36 and GHSR-1a receptors was conserved upon deletion of Lys^6^ from [azaLeu^3^]GHRP-6 (**6**), further truncation was unfruitful, and replacement of His^1^ for proline, propionic acid, glycine and D-alanine, all reduced binding affinity by ~10^3^ for the GHSR-1a receptor, but did not improve CD36 binding affinity (data not shown) [18]. 

### 2.4. Expanding Aza-GHRP-6 Analogue Diversity by azaGly Alkylation

In the synthesis of the aza-GHRP-6 analogues, the application of *N*-Fmoc-aza-amino acid chlorides in split-and-mix chemistry was initially used to prepare azapeptide libraries [18]. The activation of *N*’-alkyl 9-fluorenylmethyl carbazates with phosgene gave reactive *N*-Fmoc-aza-amino acid chlorides (e.g., **19**, Scheme 1) for introduction into peptides linked to solid support [25]. Employing resin separated into solvent permeable plastic mesh containers (IRORI Kans™), the *N*-Fmoc-aza-amino acid chlorides were reacted in parallel to synthesize azapeptides in sufficient purities and yields for biological studies. Using Fmoc-based solid-phase peptide synthesis (SPPS) and a combinatorial split-and-mix approach [26], a library of about fifty aza-GHRP-6 derivatives was prepared to examine CD36 binding affinity, selectivity and activity [18]. 

Application of *N*-Fmoc-aza-amino acid chlorides necessitated solution-phase synthesis of the *N*’-alkyl fluoren-9-ylmethyl carbazates (e.g., **18**) [25], typically by reductive amination procedures using Fmoc-hydrazide (**17**) and various carbonyl compounds. This synthetic chemistry denied access to certain aza-residues, such as aza-propargylglycine (azaPra), which would offer potential for synthesizing azapeptide “libraries from libraries” using, for example, the copper-catalyzed azide alkyne cycloaddition reaction (“click chemistry”). The installation of a broader diversity of aza-residues was subsequently achieved using a “submonomer” method featuring the alkylation and arylation of an aza-glycine semicarbazone [27], as reviewed elsewhere [28]. Strides in the discovery of selective CD36 azapeptide ligands were made employing this method to construct analogues for deciphering the structural and conformational requirements for activity. 

In an example of the “submonomer” method (Scheme 2) [27], benzaldehyde hydrazone was activated with *p*-nitrophenyl chloroformate as a carbonyl donor and coupled to the resin-bound peptide to provide aza-glycine semicarbazone **26**. The significant pKa differences of the semicarbazone, urea, amide and carbamate NH in the azaGly-peptide sequence (e.g., **26**) permitted chemoselective deprotonation and alkylation of the aza-residue semicarbazone to add various side chains, such as benzyl to give azaPhe-peptide **27**. Azapeptides (e.g., **8**) were isolated by HPLC after a sequence featuring orthogonal deprotection of the semicarbazone (e.g., **27**) using hydroxylamine hydrochloride in pyridine, amino acylation of the semicarbazide (e.g., **22**), peptide elongation, followed by resin cleavage and deprotection. Employing such methods, [azaPhe^4^]GHRP-6 derivatives were synthesized using a variety of substituted benzyl halides in the semicarbazone alkylation step and their structure–activity relationships (SAR) were examined *in vitro*. Moreover, the submonomer approach gave access to a broader diversity of azapeptides by arylation, Michael addition and conjugate addition-elimination chemistries (*vide infra*) [28].

### 2.5. Divergent Effects of Aza-GHRP-6 Analogues on Microvascular Sprouting Reveals the Relevance of azaPhe^4^ Ring Substitution

Divergent effects of aza-GHRP-6 analogues on choroidal neovascularization were observed in an *ex vivo* microvascular sprouting assay using mouse choroid explants. For example, although [azaTyr^4^]GHRP-6 (**9**) and [Ala^1^, azaPhe^4^]GHRP-6 (**15**) displayed similar receptor binding affinities and CD spectra with curve shapes indicative of β-turn conformations [18], aza-tyrosine analogue **9** exhibited anti-angiogenic activity and aza-phenylalanine analogue **15** had limited if not a slightly pro-angiogenic effect on the choroidal explants. Quantification of microvascular sprouting after 4-day treatment with [azaTyr^4^]GHRP-6 (**9**) revealed 17% choroid sprouting, validating its antiangiogenic properties compared to the control, vehicle condition which produced 53% neovascularization. On the other hand, [Ala^1^, azaPhe^4^]-GHRP-6 (**15**), which exhibited high binding affinity and CD36 selectivity, increased slightly neovascularization compared to the control, likely due to the activation of Src kinase recruitment and a vascular endothelial growth factor (VEGF)-driven protein kinase B (Akt) phosphorylation pathway [7]. Azapeptide ligands of CD36 which lacked GHS-R1a receptor binding affinity were shown to induce different biological responses, contingent upon the structure and location of the aza-amino acid residue. The divergent effects on angiogenesis were investigated further using [azaPhe^4^]-GHRP-6 analogues possessing the parent His^1^ residue, an Ala^1^ substitution, and diverse side-chain modifications at the azaPhe^4^ residue [29]. 

### 2.6. Effect of His^1^ and azaPhe^4^ Substitutions on GHRP-6 Derivative Binding Affinity and Bioactivity

Intrigued by the similar β-turn CD spectra and divergent effects on angiogenesis of lead azaPhe^4^-GHRP-6 analogues, we synthesized a library of derivatives using the submonomer approach to explore further the importance of the aromatic ring of the aza-residue [29]. Aromatic substituents were installed to study the influences of electronic density and χ-dihedral angle side chain geometry on activity (Figure 2). A subset of the substituted aza-Phe^4^ analogues were synthesized with alanine {e.g., [Ala^1^, azaPhe^4^]GHRP-6 (**15**) series **28**} in lieu of histidine {e.g., [azaPhe^4^]GHRP-6 (**8**) series **29**} to examine the influence of the *N*-terminal residue. 

Nitric oxide (NO) production by immune cells such as monocytes and macrophages has been considered an index of the inflammatory process in response to stimuli such as TLR agonists, proinflammatory cytokines and interferon γ. Azapeptide analogues were examined for their ability to reduce NO production in macrophages activated with the TLR-2 agonist (*R*)-fibroblast-stimulating lipopeptide (R-FSL-1). The physiological mediator NO is a marker of inflammation that can be readily trapped to provide a measurable fluorescent adduct [30]. The reduction of NO release from activated macrophages was considered a prerequisite indicator of CD36-mediated activity and used to select lead candidates for receptor binding affinity and angiogenesis studies in a microvascular sprouting assay using choroid explants. Among the twenty-five azaPhe analogues tested, sixteen decreased NO production and the remaining exhibited no significant effect. Notably, [azaPhe^4^]-GHRP-6 (**8**) and [Ala^1^, azaPhe^4^]-GHRP-6 (**15**) showed respectively no effect and the most significant ability to decrease NO production, but neither influenced vascular growth in the angiogenesis assay [29]. The CD36 binding affinities of the sixteen relatively similar azaPhe^4^ analogues that decreased NO production differed by about 17-fold (1.65 μM–28.2 μM); however, contingent on aromatic substituent the [azaPhe^4^]-GHRP-6 analogues exhibited divergent effects on choroidal neovascularization. 

Anti-angiogenic activity was demonstrated by [azaTyr^4^]- and [aza(4-F)Phe^4^]-GHRP-6 (**9** and **29b**). Moreover, anti-angiogenic trends were exhibited by analogues with other 4-position electronegative groups [**29d** (4-Br) and **29f** (4-MeO)]. In contrast, [aza(4-*n*-PrO)Phe^4^]-GHRP-6 (**29h**) increased neovascularization. The subtle effects of the *N*-terminal residue were indicated further by the anti- and pro-angiogenic trends exhibited respectively by [Ala^1^, aza(4-Cl)Phe^4^]-GHRP-6 (**28c**) and [aza(4-Cl)Phe^4^]-GHRP-6 (**29c**). In contrast to the relatively high binding affinity and anti-angiogenic effects of [azaTyr^4^]GHRP-6 (**9**), [aza(3-HO)Phe^4^]GHRP-6 (**29n**) exhibited about 15-fold lower CD36 binding affinity and no effect on microvascular growth. 

### 2.7. Further Probing of the azaPhe^4^ Residue Using aza-arylGly^4^- and aza-1,2,3-triazole-3-Ala^4^-GHRP-6 Analogues

Side chain diversity was expanded beyond the benzyl substituents in the [azaPhe^4^]GHRP-6 analogues by synthetic approaches that delivered respectively aza-arylglycine and aza-1-aryl-2,3-triazole-3-alanine residues (e.g., **30**–**34**). This chemistry featured respectively Cu-catalyzed *N*-arylation of azaGly semicarbazones [31], and azide-alkyne cycloadditions of aza-propargylglycine (azaPra) residues (Figure 3) [32]. In contrast to azaTrp residues which were acid labile and decomposed to azaGly peptides (*vide supra*) [25], aza-indolylglycine **31** was stable to the TFA conditions used to cleave azapeptides from the Rink amide resin. The CD36 binding affinities of the aza-arylglycine and aza-1-aryl-2,3-triazole-3-alanine GHRP-6 analogues were comparable to GHRP-6 (**1**) in the surface plasmon resonance (SPR) binding assay [22]. For example, [aza-1-aryl-2,3-triazole-3-alanine^4^]GHRP-6 analogues **33** and **34** exhibited K_d_ values around 2-10-fold lower than GHRP-6 (**1**) in the SPR binding assay (unpublished results). 

### 2.8. Exploring Potential Salt–Bridge Interactions Between CD36 and azaGlu-GHRP-6 Analogues 

On engagement of CD36, GHRP-6 analogues such as hexarelin (**38**, His-d-(2-Me)Trp-Ala-Trp-d-Phe-Lys-NH_2_) and azapeptide derivatives bind the Asn^132^–Glu^177^ sequence, which overlaps with the binding site for oxLDL [19]. In this region, three conserved lysine residues (Lys^163^, Lys^164^ and Lys^166^) contribute to binding of oxLDL [20]. Hypothesizing that the same lysine residues may participate in GHRP-6 binding by cationic-π interactions with the d-Trp^2^, Trp^4^, d-Phe^5^ aromatic residues, a series of aza-glutamate analogues were synthesized to examine the potential of salt–bridge interactions to improve receptor binding [21]. Conjugate and S_N_2’ additions of azaGly semicarbazone peptides (e.g., **26**) onto Michael acceptors and allylic acetates were respectively achieved using *tert*-butyliminotri-(pyrrolidino)phospharane (BTPP) as base to install different azaGlu analogues at the Ala^3^ (e.g., **35**) and Trp^4^ (e.g., **36** and **37**) positions of GHRP-6 (Figure 4). 

At the Ala^3^ position, comparable CD36 binding affinities to the GHRP-6 analogue standard hexarelin (**38**, 2.3 μM) were demonstrated by [aza-(cyanoethyl)Gly^3^]GHRP-6 (**35b**) and [aza-(ethyl diethylphosphoryl)Gly^3^]GHRP-6 (**35e**). In contrast, [azaGlu^3^]GHRP-6 (**35a**) and [aza-(ethyl methylphosphoryl)Gly^3^]GHRP-6 (**35c**) bound with ~10-fold lower affinity, and [aza-(ethyl dimethylphosphoryl)Gly^3^]GHRP-6 (**35d**) exhibited > 500-lower CD36 binding affinity. At the Trp^4^ position, compared to hexarelin (**38**), less significant decreases in CD36 binding affinity were exhibited by azaGlu analogues {e.g., [azaGlu^4^]GHRP-6 (**36a**, ~5-fold) and [aza-(*E*/*Z*)-(2-carboxy-*p*-nitrocinnamyl)Gly^4^]GHRP-6 (**37a**, ~8-fold)} compared to non-ionic aliphatic aza-residues: [aza-(ethyl methylphosphoryl)Gly^4^]- (**36b**), [aza-(ethyl diethylphosphoryl)Gly^4^]- (**36d**) and [aza-(*E*/*Z*)-(2-cyano-*p*-methoxycinnamyl)Gly^4^]GHRP-6 (**37b**) all showing >10^3^-fold drops in binding affinity. The relatively better binding affinity of the azaGlu^4^ analogues may be due to potential to engage in salt bridges with CD36 [21]. 

### 2.9. Aza-Lysine GHRP-6 Analogues

Azapeptides with basic side chains were pursued with an interest to replace the *C*-terminal Lys residue in GHRP-6 (Scheme 3). Aza-lysine GHRP-6 analogues were synthesized using two different submonomer procedures. Alkylation of semicarbazones (e.g., **39**) was used to install aza-amino acids with chloroalkyl and propargyl side chains [33,34,35]. Chloride **40** was employed in different S_N_2 displacement reactions with a variety of amines and sodium azide to prepare various aza-ornithine, aza-arginine and aza-lysine derivatives (e.g., **44**–**47**) [33,34]. Alternatively, aza-propargylglycine (azaPra) residues **42** were employed in copper-catalyzed A^3^-coupling reactions, between alkyne, aldehyde, and amine components, to prepare aminobutynylglycine peptides (e.g., **48**–**50**) [35]. In the A^3^-coupling, the amine and aldehyde components combine to form an imine, which is attacked by a metal acetylide nucleophile formed from the activation of the alkyne by the copper catalyst [36].

Combined, these two methods provided over thirty [azaLys]GHRP-6 analogues with various substituents on the amino group, different degrees of side chain saturation and chain length, as well as azaLys residues placed respectively at the Ala^3^, Trp^4^, d-Phe^5^, and Lys^6^ positions (Figure 5). Analysis by SPR of select members of the series revealed that [azaLys^4^]GHRP-6 analogues (e.g., **49f**–**g**,**k**–**p**) exhibited micromolar CD36 binding affinities (unpublished results). Notably, [azaLys^6^]GHRP-6 (**47a**) was found to be inactive and used subsequently as a negative control in studies of azapeptide CD36 modulator activity [29]. Moreover, [aza-(*N*,*N*-diallylaminobut-2-ynyl)Gly^4^]-GHRP-6 (**49f**) exhibited therapeutic potential in a model of atherosclerosis (*vide infra*). In addition, intramolecular variations of the A^3^-coupling reaction were developed to access cyclic azapeptide derivatives of GHRP-6 with promising results [37,38,39].

### 2.10. Unprecedented Binding Affinity and Activity Achieved by Azacyclopeptide-GHRP-6 Analogue Synthesis by A^3^-Macrocyclization

Macrocyclization has been widely employed for enhancing the metabolic stability and increasing receptor binding affinity of peptide ligands, due in part to the stabilization of active conformers [40]. Considering the relevance of an active β-turn conformer in linear azapeptide CD36 modulators such as [azaTyr^4^]GHRP-6 (**9**) [18], cyclic analogues were pursued to increase conformational rigidity and improve binding affinity by minimizing the energy of folding for target engagement [40]. Macrocyclic aza-GHRP-6 analogues were synthesized using an intramolecular variation of the A^3^-coupling reaction [37]. The cyclic aza-GHRP-6 analogues were cross-linked using formaldehyde as a linchpin in a copper-catalyzed macrocyclization between the side chains of azaPra and *N^ε^*-(alkyl)Lys residues. The so-called A^3^-macrocyclization was performed following azaPra introduction as well as after azapeptide linear sequence completion (Scheme 4). 

Different ring-sized aza-GHRP-6 macrocycles (e.g., **54**–**57**, Figure 6) were synthesized by confining the *N ^ε^*-(alkyl)Lys to the *C*-terminus and moving the azaPra residue systematically towards the *N*-terminus of the peptide sequence. Diverse azacyclopeptides (e.g., **58a**–**f**) were synthesized by A^3^-macrocyclizations on linear substrates with the *N ^ε^*-(alkyl)Lys residue in other locations, as well as with different alkyl groups on the *N^ε^*-amine. A systematic replacement of key residues in azacyclopeptide **57c** by alanine was accomplished by an approach featuring installation of the azaPra residue using the corresponding Fmoc-aza-propargylglycine acid chloride [38]. Moreover, replacement of the azaPra residue in azacyclopeptides **56** and **57c** with *R-* and *S*-propargylglycine residues was achieved using the A^3^-macrocyclization to provide configurationally stable cyclic peptides (*R*)- and (*S*)-**59** and **60** [39]. 

Biological screening demonstrated that azacyclopeptide GHRP-6 analogues **56** and **57a**–**d** and cyclic peptide counterparts (*R*)- and (*S*)-**59** and **60**, all could modulate NO overproduction in murine RAW-264.7 macrophage cells activated with R-FSL-1 as a TLR-2 agonist. Relative to linear analogues [aza-Tyr^4^]-GHRP-6 (**9**) [18] and [*N*-Ala-azaPra^1^, N^e^-(allyl)Lys^6^]-GHRP-6, which exhibited respectively similar and no influence at 10^–6^ M and 10^–7^ M, cyclic azapeptides **57b**–**d** caused significant inhibition of R-FSL-1-induced NO production at 10^–7^ M [37]. Cyclic peptides **59** and **60** reduced NO overproduction with less than half the activity as azapeptide counterparts **56** and **57c** with the (*S*)-isomers exhibiting a stronger influence than the (*R*)-counterparts [39].

Azacyclopeptide GHRP-6 analogue **57c** exhibited unprecedented CD36 binding affinity (IC_50_ 0.08 μM) in the competitive binding assay against photoactivatable [^125^I]-Tyr-Bpa-Ala-hexarelin (**3**) as radiotracer [37]. Azacyclopeptides **56**, **57b** and **57d** also exhibited relatively high CD36 binding affinities (IC_50_ 0.24 μM, 1.03 μM, 0.49 μM). Relative to their azapeptide counterparts **56** and **57c**, (*S*)-cyclic peptides (*S*)-**59** and (*S*)-**60** exhibited respectively > 6- and 8-fold reduced binding affinities, and (*R*)-cyclic peptides (*R*)-**59** and (*R*)-**60** had > 13-fold lower binding affinities. 

In sum, cyclization provided azacyclopeptide GHRP-6 analogues with greater potency in reducing TLR-2 agonist-induced NO overproduction and higher binding affinities than linear counterparts reinforcing the hypothesis of an active turn geometry. Azacyclopeptides such as **56** and **57c** are currently under investigation in *in vivo* models. Moreover, cyclic azapeptides and their cyclic peptide counterparts have served as tools for understanding the biologically active conformer as discussed in detail below. 

### 2.11. Exploration of Differences between Semicarbazides and N-Aminosulfamides Using Azasulfurylpeptides 

Azasulfurylpeptides feature an *N*-aminosulfamide residue as an amino amide surrogate with potentially greater metabolic stability. *N*-Aminosulfamide residues differ from their semicarbazide counterparts by the presence of a sulfuryl (SO_2_) group which adopts tetrahedral geometry and offers two Lewis base sites in contrast to the planar carbonyl group (CO). Compared to azapeptides in which the semicarbazide situates at the central residues of β-turns, model azasulfurylpeptides have been observed to adopt γ-turn conformers in X-ray crystallographic analyses [41]. The replacement of a semicarbazide residue by an *N*-aminosulfamide counterpart was performed to examine subtle structural and conformational differences on activity. 

A diversity-oriented synthesis of azasulfurylpeptide GHRP-6 analogues **65** was conceived to prepare a set of novel CD36 modulators (Scheme 5) [42]. Protection of the sulfamide nitrogen with an Fmoc group enabled coupling of azasulfurylglycine (AsG) tripeptide **61** onto resin. After Fmoc removal, the azasulfurylglycine (AsG) residue of resin **63** was alkylated with various arylmethyl bromides using tetraethylammonium hydroxide as base in THF. Elongation of azasulfurylpeptides **64** and resin cleavage gave four azasulfurylphenylalanine^4^-GHRP-6 analogues (e.g., **65a**–**d**) and azasulfuryl-2-naphthylalanine^4^-GHRP-6 **65e**. 

In TLR-2 agonist-stimulated macrophage cells, [(4-fluoro)azasulfurylphenylalaninyl^4^]-GHRP-6 {[(4-F)AsF^4^]-GHRP-6, **65c**} exhibited significant ability to reduce NO overproduction [42]. Moreover, in the competition binding assay against [^125^I]-Tyr-Bpa-Ala-hexarelin (**3**), [(4-F)AsF^4^]-GHRP-6 (**65c**, IC_50_ 2.53 μM) had respectively 1.3-fold lower and 2.4-fold higher CD36 binding affinities relative to the semicarbazide counterparts, [azaTyr^4^]- and [aza(4-F)Phe^4^]-GHRP-6 (**9** and **29b**). The subtle switch from a carbonyl group to a sulfuryl moiety caused relatively little change on ability to mediate a TLR-2-triggered inflammatory response and on binding affinity; however, as seen earlier with [azaPhe^4^]-GHRP-6 analogues possessing different aromatic substituents [29], the switch from an azapeptide to an azasulfurylpeptide had a significant influence on neovascularization. In the mouse choroidal explant model in which [azaTyr^4^]- and [aza(4-F)Phe^4^]-GHRP-6 (**9** and **29b**) had exhibited significant anti-angiogenic activity, azasulfuryl-GHRP-6 analogue **65c** featured no anti-angiogenic activity (Figure 7). [42]. 

### 2.12. Probing for Turn Conformations Using Aza-Proline and Aza-Pipecolic Acid Mimics

Application of aza-residues had revealed the likelihood of an active turn conformer for the biological activity of the GHRP-6 analogues [18]. Moreover, the subtle change of a semicarbazide to an *N*-aminosulfamide residue indicated that turn geometry may influence angiogenic activity [42]. Attempting to probe the turn conformation in more detail, a series of analogues were prepared using covalent constraint by way of aza-proline, aza-pipecolate, lactam and aza-lactam residues. 

Aza-proline (azaPro) and aza-pipecolic acid (azaPip) analogues were introduced into GHRP-6 in the search for selective CD36 receptor ligands [18,43]. In contrast to proline and pipecolate residues, which often exist in peptides with their *N*-terminal amide in an equilibrium between *cis*- and *trans*-isomers and a typically favored *trans* amide, electrostatic repulsion between the aza-residue nitrogen and the amide carbonyl oxygen disfavor the *trans*-isomer to afford a dominant *cis*-isomer, as observed by NMR spectroscopy and X-ray crystallography [44,45,46,47,48,49,50,51]. The combination of the covalent and electronic constraints of the azaPro and azaPip residues can induce type VI β-turn geometry [44,45,46,47,48,49,50,51]. The azaPro analogues were introduced by acylation of peptide on resin using the corresponding Fmoc protected aza-amino acid chloride **68**, which was synthesized by a route featuring a series of protecting group manipulations and alkylation of *N*-Boc-*N*’-Cbz-hydrazine (**66**) with 1,3-dibromopropane (Scheme 6) [52]. 

Systematic replacement of each residue in the d-Trp^2^-Ala^3^-Trp^4^-d-Phe^5^ region of GHRP-6 by aza-proline (azaPro scan) afforded azapeptides **70**–**73**. Among the four, only replacement of the Ala residue with azaPro gave an analogue that retained micromolar CD36 receptor binding affinity. In the competition assay against [^125^I]-Tyr-Bpa-Ala-hexarelin (**3**), [azaPro^3^]-GHRP-6 (**71**, Figure 8) exhibited CD36 binding affinity (IC_50_ 9.81 μM) [18]. In addition, azaPro^3^-peptide **71** was CD36 selective and had no binding affinity for the GHS-R1a receptor. 

The related aza-pipecolic acid derivatives were synthesized by a route featuring pericyclic chemistry on azopeptides (e.g., **76a** and **76b**, Scheme 7) [43]. Aza-pipecolate peptides **77** were prepared by coupling *N*-Fmoc-aza-glycine *N*-hydroxysuccinimide ester (**74**) to peptide resin, oxidation of the azaGly residue using *N*-bromosuccinimide (NBS) to give the corresponding azo-glycine-peptide **76**, followed by [4 + 2] cycloaddition reactions with butadiene and 2,3-dimethylbutadiene [43]. After elongation of **77** and resin cleavage with concomitant protecting group removal, ∆^4^-dehydro-azaPip derivatives **78**–**80** were isolated. Hydrogenation of **78** and **80** with palladium-on-carbon provided access to the corresponding saturated azaPip analogues **81** and **82**. After treatment with a TLR-2 agonist, four of the five [azaPip^3^]- and [azaPip^4^]GHRP-6 derivatives (**78** and **80**–**82**) decreased NO production in macrophages with activities similar to that of [azaTyr^4^]GHRP-6 (**9**). On the other hand, [(4,5-dimethyl-Δ^4^)azaPip^3^]GHRP-6 (**79**) was inactive [43] indicating that the steric bulk of the additional methyl groups interfered likely with receptor engagement. 

### 2.13. Probing for Turn Conformations Using Lactam and Aza-Lactam Residues

The application of a covalent bridge between the *N*-terminal residue side chain and the amine of the neighboring *C*-terminal residue in a lactam has been used with success in constraining peptides into rigid geometry, such as β-turn conformations, since the pioneering studies of the Merck laboratory [53,54,55]. Inspired by such studies, lactam analogues of GHRP-6 were prepared to restrict conformational mobility and increase CD36 selectivity and binding affinity. α-Amino and β-amino γ-lactams (Agl and Bgl) of both *R*- and *S*-configuration were introduced into the GHRP-6 peptide sequence by approaches featuring *N*-alkylation of peptide on resin using six- and five-membered cyclic sulfamidates **83** and **84**, respectively, followed by annulation under microwave heating (Scheme 8) [56].

Furthermore, the aza-variants of the α-amino γ-lactam, so called *N*-amino-imidazolidin-2-one (Aid) residues were introduced into GHPR-6 analogues (e.g., **91d**, Figure 9) by a sequence featuring alkylation of semicarbazone **39** with 1,2-dibromoethane, liberation of semicarbazide **90** with hydroxylamine in pyridine, peptide elongation and resin cleavage (Scheme 9) [57]. In model peptides, Aid residues have been shown by NMR spectroscopy and X-ray crystallography to adopt the *i* + 1 position of well-defined, compact β-turns possessing intramolecular ten-member hydrogen bonds within types II and II’ geometry contingent on ring puckering [58]. 

Systematic replacement of the first five amino acids of the GHRP-6 sequence with (*R*)- and (*S*)-Agl and Bgl, as well as Aid residues provided a total of twenty-six (aza-)lactam analogues [e.g., (*S*)- **87b**, (*S*)-**87c**, (*R*)-**88** and **91d**, Figure 9] [56,57]. Examination of CD36 and GHS-R1a binding affinities revealed a number of structure–activity relationships [56]. For example, except for the case of [(*S*)-Agl^2^]-GHRP-6 [(*S*)- **87b**] which retained CD36 selectivity (13.4 μM), replacements of the D- and L-Trp residues with lactams diminished significantly binding affinity especially for the ghrelin receptor, likely due to the loss of the aromatic side chain. Among the selective CD36 ligands, the notable binding affinity of [(*S*)-Agl^3^]-GHRP-6 [(*S*)-**87c**, 7.45 μM] and [(*R*)-Bgl^3^]-GHRP-6 [(*R*)- **88c**, 21.1 μM] indicated that a turn conformer about the Ala^3^ residue may favor receptor engagement, because complementary bent structures may be achieved by Agl and Bgl residues of opposite stereochemistry. Finally, replacement of D-Phe^5^ with (*S*)-Agl and (*R*)- and (*S*)-Bgl provided GHRP-6 analogues which bound selectively CD36 (21.4 μM–94.0 μM) [56]. 

In summary, the application of covalent constraint employing azaPro, azaPip, Agl, Bgl and Aid residues added support to conclude that an active turn conformer existed within the central residues of the hexapeptide. In general, replacement of Ala^3^ by these covalent constraining residues gave GHRP-6 analogues that retained some activity and binding affinity. The loss of aromatic side chains upon incorporation at the D-Trp, Trp and D-Phe residues may have likely mitigated beneficial conformational effects. 

### 2.14. Cardioprotective Effects of [Ala^1^, azaPhe^4^]-GHRP-6 (**15**)

The biomedical potential of specific azapeptide analogues has been further examined in mouse models featuring both myocardial and subretinal inflammation. Examples of these ongoing studies are presented using [azaTyr^4^]-GHRP-6 (**9**) and [Ala^1^, azaPhe^4^]-GHRP-6 (**15**). Azapeptides **9** and **15** have been respectively examined in models of ischemia-reperfusion injury and subretinal inflammation induced by exposure to blue-light.

CD36-selective aza-GHRP-6 ligands have exhibited beneficial cardioprotective effects in decreasing myocardial injury [59,60]. In an examination of the release of adiponectin promoted by CD36 ligands during myocardial ischemia and reperfusion, [Ala^1^, azaPhe^4^]-GHRP-6 (**15**) was found to exert cardioprotective effects against myocardial damage and dysfunction following transient left coronary artery ligation in mice [60]. Pretreatment of mice with azapeptide **15** reduced myocardial damage and plasma cardiac troponin I levels 48 h after reperfusion relative to untreated animals. A single dose of aza-phenylalanine analogue **15** before reperfusion resulted in reduction in injury due to myocardial ischemia and reperfusion (Figure 10). 

The effects of **15** were shown to be CD36 dependent; azapeptide treatment did not reduce injury in CD36-deficient mice. Treatment with azapeptide **15** improved myocardial function without significant change of heart rate as determined by real-time volume conductance and pressure-recording analyses. Moreover, plasma levels of adiponectin were transiently elevated and myocardial fatty acid uptake and non-esterified fatty acid mobilization were reduced by treatment with azapeptide **15**. Increases in adipose mRNA levels of PPARγ targeted genes (e.g., fatty acid transporter protein 1 (Fatp1) and phosphoenolpyruvate carboxykinase 1 (Pck1)) were observed following treatment with azapeptide **15** in CD36+/+ mice. Mice treated with azapeptide **15** and subjected to myocardial ischemia and reperfusion exhibited a small but significant reduction of mitochondrial-derived reactive oxygen species in saponin-permeabilized cardiomyocyte bundles, preservation of aconitase activity, augmented cyclooxygenase-2 (COX-2) expression and increased AMP-activated protein kinase (AMPK), Akt, and acetyl-CoA carboxylase (ACC) phosphorylation, as well as reduced cytosolic cytochrome c release and caspase 3 activity. The role of adiponectin in sustaining the cardioprotective effect of azapeptide **15** after myocardial ischemia and reperfusion was further delineated by the annulation of restorative effects in the presence of an anti-adiponectin neutralizing antibody. In sum, treatment with [Ala^1^, azaPhe^4^]-GHRP-6 (**15**) enhanced the cardioprotective effect of adiponectin in a CD36-dependent manner to prevent injury in induced murine models [60]. Selective azapeptide CD36 ligands offer beneficial means for treating myocardial ischemia and reperfusion induced injury and related cardiometabolic and immunological disorders. 

In the course of review, a publication in press was reported in which [aza-(*N*,*N*-diallylaminobut-2-ynyl)Gly^4^]-GHRP-6 (**49f**) was shown to diminish aortic lesion progression and reduce lesions in the aortic sinus of atherosclerotic mice below pre-existing levels [61]. Moreover, the effects of azapeptide **49f** were associated with a relative increase of M2-like macrophages in lesions and reduced systemic inflammation [61].

### 2.15. Mitigation of Subretinal Macrophage-Driven Inflammation Using [azaTyr^4^]GHRP-6 (**9**)

The CD36-selective modulator [azaTyr^4^]GHRP-6 (**9**) exhibited cytoprotective effects and preserved retinal function under photooxidative stress conditions simulating chronic inflammation [62]. Azapeptide **9** reduced proinflammatory cytokine levels and mononuclear phagocyte recruitment in wild-type (WT, C57BL6) mice exposed to blue LED-light for five days at an illuminance of 6000 lux (Figure 11). A mechanistic study of the modulation of the immune response was performed using fluorescent antibody for ionized calcium binding adapter molecule 1 (IBA-1) and revealed that subretinal-activated mononuclear phagocyte accumulation was reduced by azapeptide **9** in WT mice [62]. 

Azapeptide **9** mitigated the activation of the CD36-TLR-2/6 complex after induction by R-FSL as diacyl peptide TLR-2-agonist. The influence of [azaTyr^4^]GHRP-6 (**9**) on the CD36-TLR-2 complex was ascertained by measuring Förster resonance energy transfer (FRET) efficiency using antibodies for the respective proteins coupled to energy donor and acceptor Cy3 and Cy5 dyes. The TLR-2 agonist R-FSL1 induced an increase in FRET indicating a rapid association between CD36 and TLR-2 on the membranes of cultured mononuclear phagocytes. Aza-tyrosine analogue **9** attenuated the R-FSL1-induced energy transfer efficiency between Cy3 and Cy5 indicating the dissociation of the CD36-TLR-2/6 heterodimer complex at the mononuclear phagocyte cell membrane. Disruption of the heterodimer complex by azapeptide **9** resulted in alteration of the downstream signaling of the activated TLR-2 heterodimer as observed in the photo-oxidative stress-triggered pro-inflammatory cascade [62,63]. Consequently, western blot analysis indicated that treatment with **9** altered the TLR-2-signaling pathway with reduction of the phosphorylation of TLR-2-induced downstream signals (e.g., interleukin-1 receptor-associated kinase-4 (IRAK4), nuclear factor-kappa B (NF-κB), c-Jun N-terminal kinase (JNK) and the P38 mitogen-activated protein kinase (P38 MAPK)). 

As a co-receptor of TLR-2, CD36 contributes in the activation of activator protein 1 (AP-1) triggering gene transcription of proinflammatory cytokines, primarily by stimulating JNK and P38 activity. In isolated bone marrow-derived mononuclear phagocytes, which were isolated from femurs and tibias of 8-12-week-old C57BL/6 mice, azapeptide **9** decreased AP-1 activation by inhibiting JNK and P38 MAPK phosphorylation. Abrogation of inflammation was found to result in down-regulation of NF-κB and Nod-like receptor (NLR) family pyrin domain containing-3 (NLRP3) inflammasome assembly with decrease of interleukin-1β (IL-1β) secretion as observed using confocal microscopy of retinal pigment epithelium (RPE) flat mounts from illuminated CD36+/+ mice using fluorescent antibody staining. In addition, [azaTyr^4^]GHRP-6 (**9**) modulated the metabolic homeostasis of the macrophages by activating peroxisome proliferator-activated receptor-γ (PPAR-γ). Selective allosteric modulation of the interaction between CD36 and TLR-2 by azapeptide **9** decreased the immune response and reduced inflammation. Within the retina, the decreased inflammatory response caused by azapeptide **9** correlated with the preservation of neuronal cells [62], the loss of which is characteristic of retinal inflammation conditions such as retinitis pigmentosa, diabetic retinopathy and AMD (Figure 12) [63]. 

### 2.16. Implications of aza-GHRP-6 Conformation on Activity

In the development of therapeutic prototypes from linear peptides leads, dynamic fluctuation between multiple conformers of similar energy can inhibit conformational analysis. For example, GHRP-6 exhibits a CD spectrum in water characteristic of a disordered random coil with a characteristic sharp negative band around 190 nm. The application of electronic and covalent constraints can limit the conformational flexibility of the peptide to favor populations with preferred geometry. Notably, the CD curves of certain linear azapeptide CD36 modulators, such as the [aza-Phe^4^]GHRP-6 analogues (e.g., **9** and **15**), exhibit typically curve shapes indicative of β-turn conformations with negative maxima centered at around 190 and 230 nm and a positive maximum band at 215 nm [64]. Assessment of linear azapeptide GHRP-6 derivatives by CD and nuclear magnetic resonance (NMR) spectroscopy have illustrated the importance of aromatic character and β-turn secondary structure of [aza-Phe^4^]GHRP-6 derivatives for CD36 binding [18,21].

Cyclization of linear peptides has been commonly employed to provide less conformationally flexible analogues [40]. In the case of the linear peptide ligands, A^3^-macrocyclization provided respectively azacyclopeptide CD36 ligands (e.g., **56** and **57**) and their cyclic peptide counterparts (e.g., **59** and **60**) to study the conformational requirements for binding affinity, modulation of CD36-TLR-2/6 complex signaling pathways and reduction of macrophage-driven inflammation [39]. The configurationally stable (*R*)- and (*S*)-cyclic peptides were compared with their azacyclopeptide counterparts to investigate the importance of semicarbazide sp^2^ and sp^3^ hybridizations for CD36 binding affinity and downstream proinflammatory responses. Conformational analysis was performed using NMR spectroscopy and computational analysis, which revealed the conformational dynamics of the cyclic analogues and indicated an active conformer featuring a type II’ β-turn geometry for azacyclopeptide **57c** (Figure 13) [39]. The semicarbazide sp^2^-configuration provided the best representation of the aza-residue geometry in the active macrocycles with a descending preference for pseudo-sp^3^ (*S*)- and (*R*)-hybridization. In this respect, although typically less active than their respective azacyclopeptides, the (*S*)-cyclic peptides exhibited greater CD36 binding affinities, were typically more potent at reducing TLR-2 agonist induced NO production, and enhanced cholesterol efflux from macrophages more effectively than the (*R*)-counterparts [39]. The azacyclopeptide counterparts exhibited typically better CD36 binding affinity and ability to reduce NO and increase cholesterol efflux in macrophages. The active structure of azacyclopeptide **57c** was suggested to adopt a compact conformer featuring a type II’ β-turn geometry centered about the d-Trp-Ala dipeptide possessing an additional bridging hydrogen bond from the *C*-terminal amine NH and the d-Trp carbonyl oxygen [39]. Spectroscopic and computational analysis of azacyclopeptides such as **57c** have provided new food for thought for the future design of improved CD36 modulators. 

## 3. Conclusions and Future Outlook

CD36 is involved in a wide range of biological processes including the regulation of inflammation, microvascular angiogenesis, and transport of oxidized lipoproteins and phospholipids. Moreover, CD36 is implicated in microorganism and cell phagocytosis, and autophagic cell cycle events. As a clinically relevant biological marker, CD36 is associated with cardiovascular biology (e.g., atherosclerosis, angiogenesis, and ischemia-reperfusion injury), cardiac metabolism in the uptake of free fatty acids, and inflammatory related pathologies. Regulatory ligands of CD36 offer promise for various biomedical applications. Although unselective, peptide ligands related to the synthetic growth hormone releasing peptide-6 (GHRP-6) demonstrated interesting CD36 modulatory activity. Azapeptide GHRP-6 analogues bind selectively CD36 presumably due to the stabilization of β-turn conformers. The structure–activity relationship studies of CD36-selective azapeptides provided ligands exhibiting divergent effects on angiogenesis. Moreover, azapeptides can curb neovascularization, reduce nitric oxide production, and decrease immune responses mediated by mononuclear phagocytes, therefore mitigating potential neuronal injury associated with outer- and sub-retinal disorders such as retinitis pigmentosa, diabetic retinopathy and AMD. Azapeptides also exhibited cardioprotective effects and anti-atherosclerotic activity in a CD36-dependent manner. With proven beneficial activity in models of a wide range of cardiovascular, metabolic and immunological disorders, CD36-selective azapeptides are valuable for the development of peptide-based drugs. The translation of CD36 ligands into therapeutic agents is an on-going focus of our research program which is also investigating pharmacokinetic properties towards the design of optimal prototypes. Considering the challenges of conceiving drug-like ligands of receptors for which structural characterization is lacking, the synthetic methods described in this review offer pathways for the development of various peptidomimetics with selective properties for different specific indications.

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
