# Peer review of "Synthesis and Biomedical Potential of Azapeptide Modulators of the Cluster of Differentiation 36 Receptor (CD36)"

_biomedicines, 2020, doi:10.3390/biomedicines8080241_

Round 1

Reviewer 1 Report

This is an excellent summary of the work of Lubell and collaborators over the past decade or so working on azapeptides. The concept and content are certainly worth publication in Biomedicines. However, I have some concerns that should be addressed before acceptance.

1) The title is confusing. Please consider: Synthesis and Biomedical Potential of azapeptide modulators of Cluster of Differentiation 36 (CD36 would be even better)

2) My major concern with the article throughout is that is reads as a series of disconnected precis of articles. There is little connective tissue holding the various sections together: why one study follows from the previous study. Why a study was conducted? Why the substitution pattern was considered potentially useful. I am not intimately familiar with the work, but I understand that the Lubell group has contributed significantly to the generation of new methodological methods for the generation of azapeptides with various functionality, so I infer that the rationale in a lot of cases was "because we can, and it is a good model to highlight the methodology." That seems too clear in the text and yet it isn't leaned into. This gives the entire text a disjointed feeling. Rationale for why the changes are made are recommended, including potentially the inclusion of crystal structures of the receptor peptide interactions, or the computational data referred to at the end of the article.

Some notes are made to structural reasons why some substitutions were considered, but these are difficult to contextualize without structural data and images. This would likely also help tie the entire article together.

This also leads to the confusion about where the biomedical data is included. A lot of it follows the individual synthetic descriptions. This is fine, but then some is tacked on the end of the article. The bio data also doesn't seem to imply that the next study should be conducted. This makes the article challenging to read as each subheading seems completely disconnected from all the others in the review. This is by no means a terminal flaw, and some connectivity and data synthesis statements could resolve these concerns and orient the reader throughout.

The other thing to consider is I understood why the biophysics data followed synthesis, but then there was biomeidcal data sometimes after the section and sometimes at the end. This was confusing and made me flip back and forwards a lot. I think either including the biomedical data for each peptide after its synthesis or including it all as a comparative analysis at the end of the paper make sense. But dropping some in in the body, and some as the addendum makes it difficult to contextualize and compare.

3) Personal bias: I would love to see some ranges of yields for the synthetic schemes. For example Scheme 1 shows the synthesis of the Phe-like residue, but what were the yields for the transformations of 17 to 21? What residues are amenable to this strategy? Of course the reader could refer to the original article, but I really like what the authors are providing here: a detailed synthetic organic chemistry paper with extensive well conducted biophysics and biomedical data. There is little need to compromise on any of the fronts, and so more synthetic detail would be appropriate. Including yields and scope.

4) Scheme 2 recapitulates the product of Scheme 1. But the approach is predicated on the basis that Scheme 1 is not amenable to acid-sensitive chem. Could the authors provide an example where Scheme 2 works but Shceme 1 doesn't? Right now it isn't clear why one wouldn't use the simpler Scheme 1 approach for these targets. Highlighting the qualitative difference in reaction scope would be helpful and improve the rationale.

5) Nomenclature. It is confusing throughout. I strongly recommend bolding compound numbers as they keep getting confused with reference numbers (unless this is not the house style of the journal). Sometimes peptides are referred by name but not number, although the number is available. This is confusing. There are some spacing issues like in line 210 where analogue15 is written like that. This is not used in other parts of the paper.

6) A brief description of the A3 coupling reaction would be useful in line 327 to save the reader diving for google.

7) The rationale and the flow are interrupted throughout, but a representative example is provided in line 304: "[peptides] were pursued with the focus to replace the C-terminal Lys residue." This is a half-rationale and is meant as an introductory sentence, but I just sat there and thought: "why?" Why this? Why this target? What is the bio rationale? Why does its replacement stand out as important based on the previously provided biodata? I just couldn't follow. 

8) Paragraph at line 395 really confused me. We have two sets of peptides, the azaprolines and the azapipecolic acids. The first were analyzed using biophysics, the second through functional assays. But the sentences are closely connected. It took me a moment to make sense of the paragraph and realize that the two are not comparable data sets. Do all the pipecolic acids retain excellent binding affinity? Are all the prolines (except 1) inactive biologically? I don't know. I don't know if those were answered. If these aren't comparable, I'm not sure I love the discussion being in the same paragraph.

9) Ln 430 compound numbers have disappeared. I actually found this hard to follow as peptides 87, 88, and 91 are not provided. I think the structures of these would be useful to contextualize the discussion.

10) Formatting in the figures seems to be weird and have a lot of figments. Fig 9 is particularly guilty. Lots of # signs.

11) The bio discussion from Ln 481 was difficult to follow. There are a lot of connections and discussions of biological mechanisms of action which is awesome. However, there is very little detail as to how it was determined. For example, what technique was used to show that 9 induced dissociation of the CD-36-TLR-2/6 heterodimer?

12) It looks like the data in paragraph 490 was acquired in an in vivo model. This should be explicitly stated, and what type of mouse was used (infering from Fig 10).

13) The conformational discussion in paragraph of ln 514 is strong. It seems tacked on as an afterthought. I would centre this as it gives a good opportunity to tie data from the different studies together. I also wonder if affinity of the peptides could be rank ordered in a table as the data all originates from the same group and uses a limited number of techniques. I wonder if the authors could normalize the data and provide it all as corrected relative affinities. That would be amazingly useful to researchers of CD36-peptide interactions. It seems a missed opportunity.

I enjoyed reading the article, but I had to read it several times and pull up the original articles to contextualize what i was reading. I don't think this is the intent of the authors who aim to have this as a stand-alone summation of this body of work. A little more connective material, comparisons between the results of different studies, and an attempt to reorganize the text into a cohesive whole would make this an incredibly strong example of a very well conducted report of an impressive peptide medicinal chemistry campaign that would be an excellent example for students of the discipline. 

Author Response

This is an excellent summary of the work of Lubell and collaborators over the past decade or so working on azapeptides. The concept and content are certainly worth publication in Biomedicines. However, I have some concerns that should be addressed before acceptance.

Reply: We thank the referee for the comments and critical feedback and have made efforts described below to address the raised points.

1) The title is confusing. Please consider: Synthesis and Biomedical Potential of azapeptide modulators of Cluster of Differentiation 36 (CD36 would be even better)

As requested: the title now reads “Synthesis and Biomedical Potential of Azapeptide Modulators of the Cluster of Differentiation 36 Receptor (CD36)”

2) My major concern with the article throughout is that is reads as a series of disconnected precis of articles. There is little connective tissue holding the various sections together: why one study follows from the previous study. Why a study was conducted? Why the substitution pattern was considered potentially useful. I am not intimately familiar with the work, but I understand that the Lubell group has contributed significantly to the generation of new methodological methods for the generation of azapeptides with various functionality, so I infer that the rationale in a lot of cases was "because we can, and it is a good model to highlight the methodology." That seems too clear in the text and yet it isn't leaned into. This gives the entire text a disjointed feeling. Rationale for why the changes are made are recommended, including potentially the inclusion of crystal structures of the receptor peptide interactions, or the computational data referred to at the end of the article.

As requested: Efforts have been made to better contextualize the research. For example, in the introduction a phrase was added to note that there are no “inclusion of crystal structures of the receptor peptide” complexes, nor of the receptor alone at the moment: “In the absence of structural data for CD36 alone nor in complex with small molecules, lead optimization relied on structure-­activity relationship data obtained from different assays designed to assess the pleiotropic effects of the receptor.” Another phrase was added to provide the logic for using azapeptides: “Peptide 1 has been characterized to adopt a random coil conformation in solution based on circular dichroism spectroscopic analysis (vide infra) [18], in spite the central D-­Trp-­L-­Ala-­L-­Trp-­D-­Phe sequence possessing alternating L-­ and D-­amino acid pairs, which are commonly found at the central resides of b-­turns [23]. Considering that a preferred turn geometry may be adopted on receptor binding, GHRP-­6 analogues that discriminate between CD36 and the GHS-­R1a receptor were pursued by employing semicarbazide residues as amino amide surrogates to favor such conformers and improve selective binding affinity.” Similarly, phrases were added to explain the logic in making cyclic analogs: “Considering the relevance of an active b-­turn conformer in linear azapeptide CD36 modulators such as [azaTyr4]GHRP-­6 (9) [18], cyclic analogs were pursued to increase conformational rigidity and improve binding affinity by minimizing the energy of folding for target engagement [40].” Phrases were also added to better introduce the study of azasulfurylpeptides: “N-­Aminosulfamide residues differ from their semicarbazide counterparts by the presence of a sulfuryl (SO2) group which adopts tetrahedral geometry and offers two Lewis base sites in contrast to the planar carbonyl group (CO). Compared to azapeptides in which the semicarbazide situates at the central residues of b-­turns, model azasulfurylpeptides have been observed to adopt g-­turn conformers in X-­ray crystallographic analyses [41].
The replacement of a semicarbazide residue by the N-­aminosulfamide counterpart was thus performed to examine subtle structural and conformational differences on activity.” Moreover, phrases were added to introduce the use of covalent constraint: “Application of aza-­residues had revealed the likelihood of an active turn conformer for the biological activity of the GHRP-­6 analogs [18]. Moreover, the subtle change of a semicarbazide to an N-­aminosulfamide residue indicated that turn geometry may influence angiogenic activity [42]. Probing the turn  conformation in more detail, a series of analogs were prepared using covalent constraint by way of aza-­proline, aza-­pipecolate, lactam and aza-­lactam residues.”

Some notes are made to structural reasons why some substitutions were considered, but these are difficult to contextualize without structural data and images. This would likely also help tie the entire article together.

Reply: As mentioned, there are no X-­ray nor NMR structures of CD36 alone or in complexation with receptors. The importance of using a spectrum of synthetic analogs to gain insight into structure-­activity relationships has thus been highlighted. More contextualization for the application of the different synthetic methods has now been added.

This also leads to the confusion about where the biomedical data is included. A lot of it follows the individual synthetic descriptions. This is fine, but then some is tacked on the end of the article. The bio data also doesn't seem to imply that the next study should be conducted. This makes the article challenging to read as each subheading seems completely disconnected from all the others in the review.
This is by no means a terminal flaw, and some connectivity and data synthesis statements could resolve these concerns and orient the reader throughout.

The other thing to consider is I understood why the biophysics data followed synthesis, but then there was biomeidcal data sometimes after the section and sometimes at the end. This was confusing and made me flip back and forwards a lot. I think either including the biomedical data for each peptide after its synthesis or including it all as a comparative analysis at the end of the paper make sense. But dropping some in in the body, and some as the addendum makes it difficult to contextualize and compare.

Reply: The biological data has been added after specific sections to highlight the influence of the synthetic modification on binding and specific activities. In the latter sections, a deeper analysis of two lead peptides is provided with in vivo data which supports application of the azapeptides for specific indications such as cardiovascular disease and macrophage-­driven inflammation. As mentioned above, additional text has been added to better clarify the former sections. To introduce the latter sections, the following text was added: “The biomedical potential of specific azapeptide analogues has been further examined in mouse models of cardiovascular and inflammatory conditions. Examples of these ongoing studies are presented using [azaTyr4]-­GHRP-­6 (9) and [Ala1, azaPhe4]-­GHRP-­6 (15). Azapeptides 9 and 15 have been respectively examined in models of retinal inflammation and myocardial injury.”

3) Personal bias: I would love to see some ranges of yields for the synthetic schemes. For example Scheme 1 shows the synthesis of the Phe-­like residue, but what were the yields for the transformations of 17 to 21? What residues are amenable to this strategy? Of course the reader could refer to the original article, but I really like what the authors are providing here: a detailed synthetic organic chemistry paper with extensive well conducted biophysics and biomedical data. There is little need to compromise on any of the fronts, and so more synthetic detail would be appropriate. Including yields and scope.

Reply: the manuscript focus is on the medicinal chemistry and its application to study structure-­activity relationships. As mentioned in the text, adequate amounts of material in suitable purity and yield were provided for biological analyses. Moreover, a review on the synthetic methods for making azapeptides was published in Acc. Chem. Res. in 2017 and cited in reference 28. Although the latter does not contain the azasulfurylpeptide chemistry nor more recent published methods, it is a useful starting point for assessing the importance of yields. In the current work, these have not been added.

4) Scheme 2 recapitulates the product of Scheme 1. But the approach is predicated on the basis that Scheme 1 is not amenable to acid-­sensitive chem. Could the authors provide an example where Scheme 2 works but Shceme 1 doesn't? Right now it isn't clear why one wouldn't use the simpler Scheme 1 approach for these targets. Highlighting the qualitative difference in reaction scope would be helpful and improve the rationale.

Reply: Schemes 1 and 2 give head-­to-­head comparisons of two synthetic methods to highlight their similarities and differences. The corresponding text in the section provides an explanation for the utility of the method in Scheme 2 for obtaining greater structural diversity. The title of the section was changed to further illustrate this point.

5) Nomenclature. It is confusing throughout. I strongly recommend bolding compound numbers as they keep getting confused with reference numbers (unless this is not the house style of the journal). Sometimes peptides are referred by name but not number, although the number is available. This is confusing. There are some spacing issues like in line 210 where analogue15 is written like that. This is not used in other parts of the paper.

As requested: The authors are in complete agreement and disappointed that the standard bold compound-­numbering used in the manuscript was stripped in the processing of the document. The bold compound-­numbering has been re-­added at a significant cost in time to the authors, who implore the editorial staff to be more mindful to maintain such formalism in further processing and in other chemistry-­oriented manuscripts in the future. The spacing issues were corrected.

6) A brief description of the A3 coupling reaction would be useful in line 327 to save the reader diving for google.

As requested: the description and original reference were added.

7) The rationale and the flow are interrupted throughout, but a representative example is provided in line 304: "[peptides] were pursued with the focus to replace the C-­terminal Lys residue." This is a half-­rationale and is meant as an introductory sentence, but I just sat there and thought: "why?" Why this?
Why this target? What is the bio rationale? Why does its replacement stand out as important based on the previously provided biodata? I just couldn't follow.

Reply: Admittedly, the exploration of azaLys surrogates was a more “wild haired” experiment, which was performed as a try and see at the time. In addition to novel methods for making aza-­residues, three important results did arrive from this effort: 1) azaLys6-­GHRP-­6 became a valuable negative control, 2) the intermolecular A3-­reaction paved the way to the A3-­macrocyclization which has provided analogs with unprecedented binding affinity and activity, 3) certain analogs did exhibit notable activity and are currently under investigation and will be reported in due time.

8) Paragraph at line 395 really confused me. We have two sets of peptides, the azaprolines and the
azapipecolic acids. The first were analyzed using biophysics, the second through functional assays. But
the sentences are closely connected. It took me a moment to make sense of the paragraph and realize
that the two are not comparable data sets. Do all the pipecolic acids retain excellent binding affinity? Are
all the prolines (except 1) inactive biologically? I don't know. I don't know if those were answered. If these
aren't comparable, I'm not sure I love the discussion being in the same paragraph.

As requested, the discussion has been separated and contextualized.

9) Ln 430 compound numbers have disappeared. I actually found this hard to follow as peptides 87,
88, and 91 are not provided. I think the structures of these would be useful to contextualize the
discussion.

Reply: a new figure and compound numbers have been added to “contextualize the discussion.”

10) Formatting in the figures seems to be weird and have a lot of figments. Fig 9 is particularly guilty.
Lots of # signs.

Reply: This appears to be a computer compatibility issue.

11) The bio discussion from Ln 481 was difficult to follow. There are a lot of connections and
discussions of biological mechanisms of action which is awesome. However, there is very little detail as to
how it was determined. For example, what technique was used to show that 9 induced dissociation of the
CD-­36-­TLR-­2/6 heterodimer?

As requested: the text now reads: “The influence of [azaTyr4]GHRP-­6 (9) on the CD36-­TLR-­2 complex
was ascertained by measuring Förster resonance energy transfer (FRET) efficiency using antibodies for
the respective proteins coupled to energy donor and acceptor Cy3 and Cy5 dyes. The TLR-­2 agonist R-­
FSL1 induced an increase in FRET indicating a rapid association between CD36 and TLR-­2 on the
membranes of cultured mononuclear phagocytes. Aza-­tyrosine analogue 9 attenuated the R-­FSL1-­
induced energy transfer efficiency between Cy3 and Cy5 indicating the dissociation of the CD36-­TLR-­2/6
heterodimer complex at the mononuclear phagocyte cell membrane.” In addition, more detail has been
added to this section to address the referee comments.

12) It looks like the data in paragraph 490 was acquired in an in vivo model. This should be explicitly
stated, and what type of mouse was used (infering from Fig 10).

As requested: the text now reads: “In bone marrow-­derived mononuclear phagocytes, which were
isolated from femurs and tibias of 8-­12-­week-­old C57BL/6 mice, azapeptide 9 decreased AP-­1 activation
by inhibiting JNK and P38 MAPK phosphorylation.” The Figure already read “WT (C57BL6) mice”.

13) The conformational discussion in paragraph of ln 514 is strong. It seems tacked on as an
afterthought. I would centre this as it gives a good opportunity to tie data from the different studies
together. I also wonder if affinity of the peptides could be rank ordered in a table as the data all originates
from the same group and uses a limited number of techniques. I wonder if the authors could normalize
the data and provide it all as corrected relative affinities. That would be amazingly useful to researchers of
CD36-­peptide interactions. It seems a missed opportunity.

Reply: The discussion of conformation runs throughout the manuscript as part of the design concepts.
The final analysis of conformation in said paragraph(s) brings the discussion to a close and provides the
most recent analyses into perspective. An additional figure has been added to strengthen this section,
which has been better contextualized, and which serves in part to indicate where the future lies.

I enjoyed reading the article, but I had to read it several times and pull up the original articles to
contextualize what i was reading. I don't think this is the intent of the authors who aim to have this as a
stand-­alone summation of this body of work. A little more connective material, comparisons between the
results of different studies, and an attempt to reorganize the text into a cohesive whole would make this
an incredibly strong example of a very well conducted report of an impressive peptide medicinal
chemistry campaign that would be an excellent example for students of the discipline.

Reply: Ideally, the new version is even more enjoyable.

Reviewer 2 Report

This review exhaustively covers more than a decade of development and pharmacological and biological investigations  of modulators of the cluster of differentiation 36 receptor , based on azapeptide analogues of GHRP-6.

 In particular, this review focuses on the chemistry for SAR studies for better understanding of the topological requirements for binding and regulating CD36 activity.

The issue of enzymatic /metabolic stability has been brushed  only superficially. On the other hand, this is a fundamental issue in peptidomimetic field. For those compounds which showed  in vivo efficacy, stability seems warranted; however, this does not consent to compare how the different modifications impact stability.

Minor concerns

Check for correct spacing between words in lines 170 and 204 and 209 and 291

Check and correct the labels of Figure 6 and Figure 9 and Figure 10

Line 378, it is said that “azaPro and azaPip can favor type VI beta-turn conformers due to a combination of 379 covalent and electronic constraints..”. Please resume also here the cis-trans isomerism explicitly; this was originally discussed in Ref [43].

Author Response

This review exhaustively covers more than a decade of development and pharmacological and biological
investigations of modulators of the cluster of differentiation 36 receptor , based on azapeptide analogues
of GHRP-­6.
In particular, this review focuses on the chemistry for SAR studies for better understanding of the
topological requirements for binding and regulating CD36 activity.
The issue of enzymatic /metabolic stability has been brushed only superficially. On the other hand, this is
a fundamental issue in peptidomimetic field. For those compounds which showed in vivo efficacy,
stability seems warranted;; however, this does not consent to compare how the different modifications
impact stability.

Reply: We recognize the important points of the reviewer regarding stability and pharmacokinetic
properties. Current research is oriented in this direction yet premature for publication. In this light the
phrase was added: “Translation of CD36 ligands into therapeutic agents is an on-­going focus of our
research program which is also investigating pharmacokinetic properties towards the design optimal
prototypes.”

Minor concerns
Check for correct spacing between words in lines 170 and 204 and 209 and 291

As requested: spaces were added.

Check and correct the labels of Figure 6 and Figure 9 and Figure 10

As mentioned, this appears to be a computer compatibility issue.

Line 378, it is said that “azaPro and azaPip can favor type VI beta-­turn conformers due to a combination
of 379 covalent and electronic constraints..”. Please resume also here the cis-­trans isomerism explicitly;;
this was originally discussed in Ref [43].

As requested: an explanation of the influences of azaPro and azaPip on amide isomer equilibrium has been added.

Round 2

Reviewer 1 Report

I would like to thank the authors for engaging with the comments.

There are only a few points I wish to return to:

4) The authors state that the point of this article is not to provide details of reaction scope or yield. I can accept this. The authors then state in point 4 that Scheme 1 and 2 provide a head-to-head comparison. Head to head comparisons normally make sense in terms of relative efficiency of processes. This isn't discussed. Instead, the article does state that Scheme 2 allows access to molecules unavailable in Scheme 1. I think the authors can have either/or: either a comparison of efficiency or a discussion of the different molecules available. But claiming that this is a comparison of efficiency without discussing efficiency is not tenable.

7) I agree with the authors! The choice is clearly, because it is is possible. But some semblance of rationale, or a discussion of the impetus for this study, or an introductory comment would be helpful. Perhaps a comment about how it might interact with a residue on the protein, or the importance is unknown, or the geometric balance could be better instituted etc. Or even: "Modifying lysine residues is a common method in peptide modification" https://chemistry-europe.onlinelibrary.wiley.com/doi/abs/10.1002/ejoc.201701174.

Author Response

4) The authors state that the point of this article is not to provide details of reaction scope or yield. I can accept this. The authors then state in point 4 that Scheme 1 and 2 provide a head-tohead comparison. Head to head comparisons normally make sense in terms of relative efficiency of processes. This isn't discussed. Instead, the article does state that Scheme 2 allows access to molecules unavailable in Scheme 1. I think the authors can have either/or: either a comparison of efficiency or a discussion of the different molecules available. But claiming that this is a comparison of efficiency without discussing efficiency is not tenable.

Previous Reply: Schemes 1 and 2 give head-to-head comparisons of two synthetic methods to highlight their similarities and differences. The corresponding text in the section provides an explanation for the utility of the method in Scheme 2 for obtaining greater structural diversity.The title of the section was changed to further illustrate this point.

Current reply, the term “head to head comparison” was never used in the manuscript. The schemes do “highlight their similarities and differences”; moreover, “a discussion of the different molecules available” is presented in the text which stated: The installation of a broader diversity of aza-residues was subsequently achieved using a “submonomer” method featuring the alkylation and arylation of an aza-glycine semicarbazone [27], as reviewed elsewhere [28]. Moreover, the submonomer approach gave access to a broader diversity of azapeptides by arylation, Michael addition and conjugate addition-elimination chemistries [28].To further clarify the point the term “(vide infra)” was added to the latter statement.

7) I agree with the authors! The choice is clearly, because it is is possible. But some semblance

of rationale, or a discussion of the impetus for this study, or an introductory comment would be

helpful. Perhaps a comment about how it might interact with a residue on the protein, or the

importance is unknown, or the geometric balance could be better instituted etc. Or even:

"Modifying lysine residues is a common method in peptide modification" https://chemistryeurope.

onlinelibrary.wiley.com/doi/abs/10.1002/ejoc.201701174.

Previous Reply: Admittedly, the exploration of azaLys surrogates was a more “wild haired”

experiment, which was performed as a try and see at the time. In addition to novel methods for

making aza-residues, three important results did arrive from this effort: 1) azaLys6-GHRP-6

became a valuable negative control, 2) the intermolecular A3-reaction paved the way to the A3-

macrocyclization which has provided analogs with unprecedented binding affinity and activity,

3) certain analogs did exhibit notable activity and are currently under investigation and will be

reported in due time.

Current reply, we believe that the initial phrase is sufficient lead in to the section: “Azapeptides

with basic side chains were pursued with the focus to replace the C-terminal Lys residue in

GHRP-6 (Figure 4).” Instead of further elaborating a justification, we have added a Note in

Proof at the end of the manuscript in which the importance of the aza-lysine analogs has been

validated in vivo:

Note Added in Proof. In the course of review, a publication in press was reported in which [aza-

(N,N-diallylaminobut-2-ynyl)Gly4]-GHRP-6 (49f) was shown to diminish aortic lesion

progression and reduce, below pre-existing levels, lesions in the aortic sinus of atherosclerotic

mice [64]. The effects of azapeptide 49f were associated with a relative increase of M2-like

macrophages in lesions and reduced systemic inflammation [64].”

Reviewer 2 Report

The authors addressed all reviewers' comments in full; to be published in the present form

Author Response

As per comments:

"The authors addressed all reviewers' comments in full; to be published in the present form."

We thank the reviewer for the efforts.